# The contribution of abortive infection to preventing populations of *Lactococcus lactis* from succumbing to infections with bacteriophage

Eduardo Rodríguez-Román[‡], Joshua A. Manuel [ID][‡], David Goldberg, Bruce R. Levin [ID]*

Department of Biology, Emory University, Atlanta, GA, United States of America

‡ ERR and JAM are co-first authors on this work.
* blevin@emory.edu

**Data Availability Statement:** All relevant data are within the manuscript and its Supporting Information files.

## Abstract

In the dairy industry bacteriophage (phage) contamination significantly impairs the production and quality of products like yogurt and cheese. To combat this issue, the strains of bacteria used as starter cultures possess mechanisms that make them resistant to phage infection, such as envelope resistance, or processes that render them immune to phage infection, such as restriction-modification and CRISPR-Cas. *Lactococcus lactis*, used to manufacture cheese and other dairy products, can also block the reproduction of infecting phages by abortive infection (Abi), a process in which phage-infected cells die before the phage replicate. We employ mathematical-computer simulation models and experiments with two *Lactococcus lactis* strains and two lytic phages to investigate the conditions under which Abi can limit the proliferation of phages in *L. lactis* populations and prevent the extinction of their populations by these viruses. According to our model, if Abi is almost perfect and there are no other populations of bacteria capable of supporting the replication of the *L. lactis* phages, Abi can protect bacterial populations from succumbing to infections with these viruses. This prediction is supported by the results of our experiment, which indicate that Abi can help protect *L. lactis* populations from extinction by lytic phage infections. However, our results also predict abortive infection is only one element of *L. lactis* defenses against phage infection. Mutant phages that can circumvent the Abi systems of these bacteria emerge. The survival of *L. lactis* populations then depends on the evolution of envelope mutants that are resistant to the evolved host-range phage.

## Introduction

In recent years, there has been a revival of interest in the practical application of bacteriophages (phages) for treating bacterial diseases in people, domestic animals, and companion animals, as well as for regulating bacterial growth in agriculture [1–3]. In industries where bacteria play a fundamental role in the manufacturing of goods, phages are viewed instead as a

**Funding:** Funds for this research were provided by a grant from the US National Institutes of GeneralMedical Sciences (https://www.nigms.nih.gov/), R35GM136407 (BRL). The funders had no role in study design, data collection and analysis, decision to publish, or preparation of the manuscript.

**Competing interests:** The authors have declared that no competing interests exist.

pest. This is especially true in the dairy industry. As a result of phage contamination, production rates and the quality of dairy products such as yogurt and cheese can deteriorate significantly, resulting in severe financial losses [4–6]. To overcome this issue, much research has focused on the exploration and the development of mechanisms to prevent bacteria from succumbing to phage infection. One such mechanism is abortive infection (Abi). The majority of research on Abi, in which phage-infected bacteria prevent the phage from replicating by induced cell death, has been conducted on *Lactococcus lactis*, bacteria commonly used in the dairy industry [4, 7, 8]. In contrast to other phage defense mechanisms such as envelope resistance, restriction-modification, and CRISPR-Cas, the expression of Abi may protect bacterial populations but not the individual bacteria exhibiting this trait [9–13].

Abi's method of action, genetics, and molecular biology have been the subject of a great deal of sophisticated research, and we know a great deal about these aspects of Abi, notably those encoded for by *L. lactis* [14–20]. Less is known about the population dynamics of Abi and how this this mechanism prevents phage infection in populations of these bacteria. Berry-hill et al. [21] found that the emergence of envelope resistance was a key contributor to bacterial survival of phage infection in populations of *Escherichia coli* with Abi. In this study, we employ a mathematical and computer simulations model to investigate the conditions under which Abi protects bacterial communities from phage infection and as a framework to facilitate the design and interpretation of experiments. We conduct *in vitro* studies with *Lactococcus lactis* and two of its lytic phages, p2 (group 936, genus *Skunavirus*) and P335 (group P335), both of them of the family *Siphoviridae*, which are commonly found in the dairy industry [22, 23], to estimate the parameters of these models and test the hypotheses derived from our examination of their properties. We focus our investigation on AbiZ, an abortive infection mechanism discovered and described by Durmaz et al. [18] as leading to premature cell lysis during phage infection, and one of the most understood Abi mechanisms on *L. lactis*. Our findings show that Abi may shield populations of *L. lactis* from phage infection under certain conditions, but this protection is transient and only one defensive element in populations of *L. lactis* to prevent their extinction by phage. Phage mutants that evade Abi are generated and ascend, and *L. lactis* mutants with envelope resistance to the phage evolve and become the dominant bacterial population.

## Results

### Short-term population dynamics: Abortive infection in *L. lactis* and its phages

We initiate our study of the contribution of Abi to protecting populations of *L. lactis* from succumbing to phage infection with short-term experiments using AbiZ⁻ (NCK4 and IL6) and AbiZ⁺ (NCK5 and IL7) bacteria and p2 and P335 phages (Fig 1). In the absence of these phages, the bacteria consume the resource, and their populations grow exponentially until the nutrients needed for replication are depleted and reach their maximum density. When combined with phage, the densities of the AbiZ⁻ population decrease and phage densities increase (Fig 1A and 1B). When AbiZ⁺ cells are mixed with phage, different dynamics are observed. In the experiment with AbiZ⁺ NCK5 and p2 (Fig 1C), the density of bacteria increases at a rate similar to that in the phage-free controls but declines slightly as the density of total p2 ascends. We attribute the increase in the density of total phages to the selection and ascent of mutant phages that can evade AbiZ. The existence of mutant p2 and P335 that evade abortive infection is anticipated from the study by Durmaz et al. [18]. The number of escape mutants was quantified by plating phage on lawns of AbiZ⁺ cells and single plaques were counted giving us evolved phage densities, p2ev and P335ev. As estimated on the AbiZ⁻ (NCK4) lawn the density

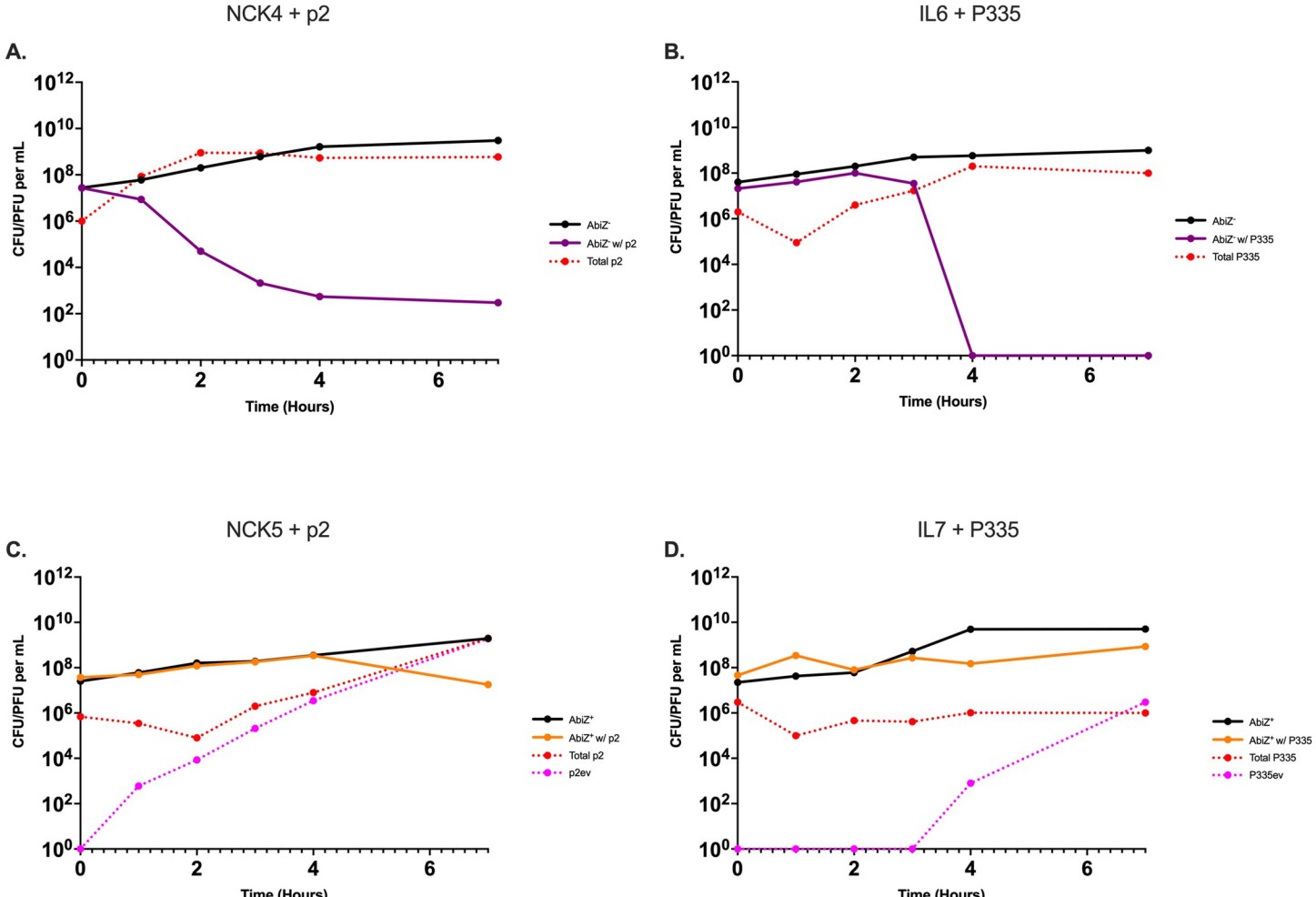

**Fig 1. Short-term population dynamics of *L. lactis* and phage.** Changes in the densities of bacteria and phage over seven hours. Solid black line represents AbiZ⁻ or AbiZ⁺ controls grown without phage present. **A:** AbiZ⁻ (NCK4) bacteria (purple) with phage p2 (dashed red). **B:** AbiZ⁻ (IL6) cells (purple) with phage P335 (dashed red). **C:** AbiZ⁺ (NCK5) cells (orange) with phage p2 (dashed red) and p2ev (dashed pink). **D:** AbiZ+ (IL7) cells (orange) with phage P335 (dashed red) and P335ev (dashed pink).

of phages, i.e. the number of plaques formed, is initially constant and then increases. As estimated on the AbiZ⁺ (NCK5) lawn the density of phage continually increases. In the corresponding experiments with the IL7 AbiZ⁺ bacteria and phage P335, the bacterial density increases at a rate lower than the controls without phage and, as estimated on the AbiZ⁻ (IL6) lawns the density of P335 is held at a level similar to that inoculated. However, as estimated on the AbiZ⁺ (IL7) lawn, the density of the P335ev Abi escape mutant increases steadily and ultimately leads to the rise in total P335 phages after 4 hours.

Two lines of evidence further support the hypothesis that the evolution of Abi-evading phage mutants accounts for the ascent of total phages able to replicate on AbiZ⁺ cells observed in Fig 1C and 1D. One is sequence data presented in the supplemental material (Table S1 in S1 File) in which a missense mutation in the major capsid protein of both phages is observed, a mutation commonly associated with Abi phage escape mutants (escapers) of other Abi systems in *L. lactis* [24, 25]. The other evidence is illustrated in S1C and S1D Fig in S1 File, where we start the experiments with AbiZ⁺ bacteria with the evolved p2ev and P335ev phage mutants

that evade abortive infection mediated by AbiZ. These phages are able to replicate on the AbiZ$^+$ bacteria and the dynamics observed is similar to that of the AbiZ$^-$ bacteria with the ancestral phage (Fig 1A and 1B), as well as the evolved phage (S1A and S1B Fig in S1 File). These short-term population dynamics are consistent for all conditions in two additional biological replicas found in S2 Fig in S1 File with the exception of one replica of IL7 and P335 in which P335ev ascended to a high enough density to drop bacterial density below our limit of detection (1x10$^1$ CFU/mL) within 7 hours.

## A mathematical model of abortive infection

To identify and evaluate the role of the different parameters that govern the population and evolutionary dynamic of interaction between phage and bacteria with an Abi system and generate hypotheses and interpret the results of our experiments, we use a mathematical model. This model is an extension of that employed in Berryhill et al. [21] to allow for evolved phage which bypass Abi. Our extended model is diagramed in Fig 2. There are four populations of bacteria, Abi negative cells (Abi$^-$) sensitive to the phage, Abi$^-$ cells resistant (refractory) to the phage, Abi positive (Abi$^+$) cells that are sensitive to the phage but have an abortive infection system, and Abi$^+$ cells envelope resistant to the phage, with designations and densities (cells per ml), $N$, $Nr$, $A$ and $Ar$, respectively. There are two populations of phage, one sensitive to abortive infection and one unaffected by Abi, respectively $P$ and $Pe$ phage per mL. $P$ and $Pe$ adsorb to the sensitive Abi$^+$ and Abi$^-$, $A$ and $N$, bacteria with a rate constant, $\delta$ (mL per cell per hour) [26]. Phage adsorption is a mass-action process occurring at a rate equal to the product of the densities of the phages, bacteria and $\delta$. The phage, $P$ and $Pe$, do not adsorb to the resistant $Nr$ and $Ar$ populations. The phages $P$ and $Pe$ that adsorb to the $N$ population produce $\beta$

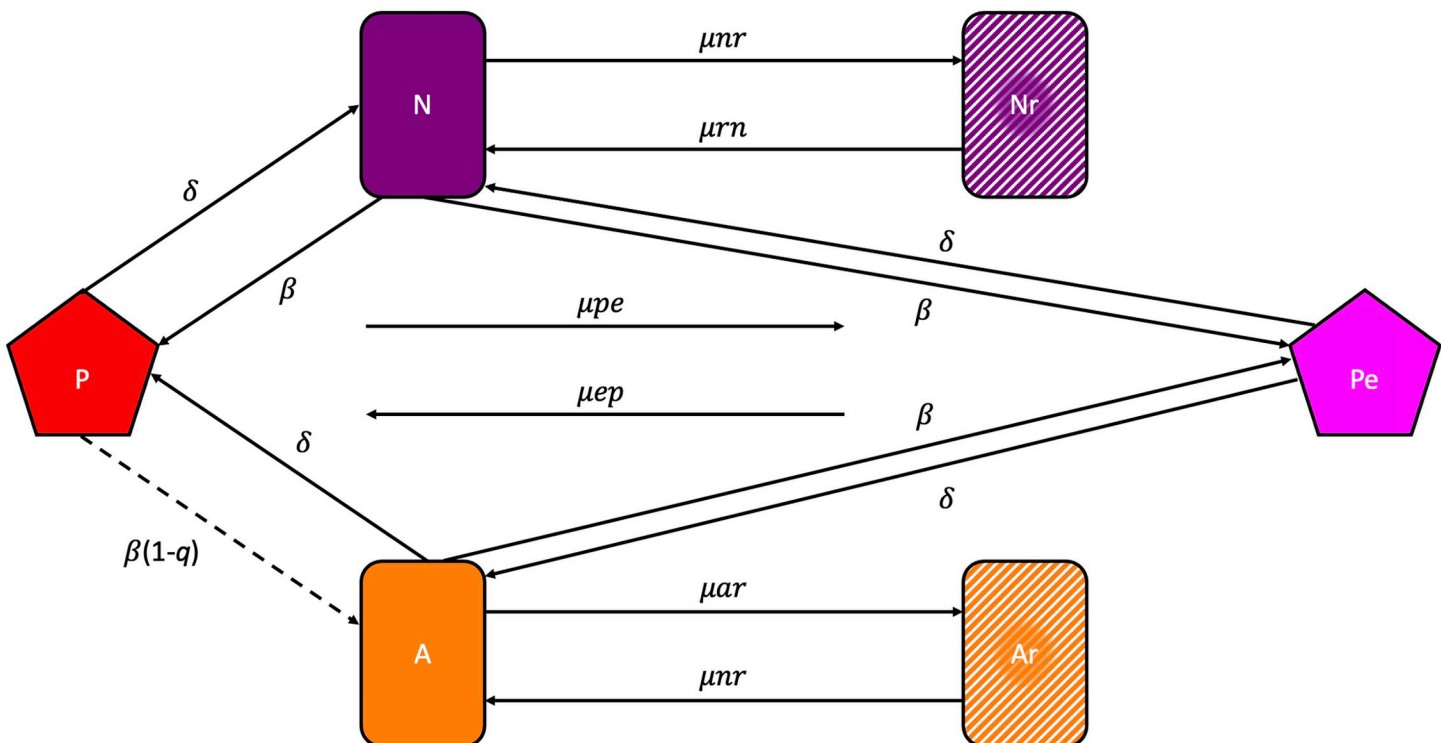

**Fig 2. Mass action model of the population and evolutionary dynamics of lytic phage and bacteria with abortive infection.** See the text and Table 1 for the definitions and dimensions of the variables and parameters.

phage particles, while *Pe* adsorbing to *A* also produces *β* phage. The parameter *q* is a measure of the efficacy of abortive infection. With a probability *q*, ($0 \leq q \leq 1$), phage *P* and the Abi$^+$ cell to which it adsorbs is lost. The remaining (1-*q*) of these infections produce *β* phage particles.

The bacterial populations grow at maximum rates, denoted as *vn*, *vnr*, *va*, *var* per cell per hour, respectively for *N*, *Nr*, *A*, and *Ar*. A limiting resource at a concentration *r* μg/mL is consumed at a rate equal to the product of parameter *e* μg/cell [27], the sum of the product of the densities of viable cells, their maximum growth rates, and a hyperbolic function, $\psi(r) = \frac{r}{r+k}$ where k μg/mL, the Monod constant [28] is the concentration of the limiting resource when the growth rate is half its maximum value. These populations transition between *A*➤*Ar*, *Ar*➤*A*, *N*➤*Nr* and *Nr*➤*N*, *P*➤*Pe* and *Pe*➤*P* at mutation rates *μar*, *μra*, *μnr*, *μrn*, *μpe*, *μep* respectively.

With these definitions and assumptions, the rates of change in the densities of bacteria and phage and concentration of the limiting resource are given by the below set of coupled differential equations.

$$\frac{dr}{dt} = -\psi(r) \cdot e \cdot (vn \cdot N + vnr \cdot Nr + va \cdot A + var \cdot Ar) \tag{1}$$

$$\frac{dN}{dt} = \psi(r) \cdot (vn \cdot N - \delta \cdot N \cdot (P + Pe) + \mu rn \cdot Nr - \mu nr \cdot N) \tag{2}$$

$$\frac{dNr}{dt} = \psi(r) \cdot (vnr \cdot Nr - \mu rn \cdot Nr + \mu nr \cdot N) \tag{3}$$

$$\frac{dA}{dt} = \psi(r) \cdot (va \cdot A - \delta \cdot A \cdot P - \delta \cdot A \cdot Pe + \mu ra \cdot Ar - \mu ar \cdot A) \tag{4}$$

$$\frac{dAr}{dt} = \psi(r) \cdot (var \cdot Ar - \mu ra \cdot Ar + \mu ar \cdot A) \tag{5}$$

$$\frac{dP}{dt} = \psi(r) \cdot (\delta \cdot N \cdot P \cdot \beta + \delta \cdot A \cdot P \cdot (1 - q) \cdot \beta - \delta \cdot A \cdot P \cdot q + \mu ep \cdot Pe - \mu pe \cdot P) \tag{6}$$

$$\frac{dPe}{dt} = \psi(r) \cdot (\delta \cdot (N + A) \cdot Pe \cdot \beta - \mu ep \cdot Pe + \mu pe \cdot P) \tag{7}$$

$$\psi(r) = \frac{r}{r + k} \tag{8}$$

## Twenty-four-hour changes in the densities of *L. lactis* and phage: Simulation and experimental results

To better understand the conditions under which Abi will protect populations of *L. lactis* from succumbing to phage infection, we consider changes in the densities of bacteria and phage over 24 hours (time 0 and time 24), anticipated from our simulations and observed empirically with the two experimental groups: *L. lactis* NCK4 AbiZ$^-$, NCK5 AbiZ$^+$ with the phage p2, and IL6 AbiZ$^-$ and IL7 AbiZ$^+$ with the phage P335. The parameters used for our simulations are derived from those estimated for these bacteria and phages (Table 1).

**Table 1. Variables and parameters.**

| Variable | Definition | Simulation Value (Dimensions) | | |
|---|---|---|---|---|
| $r$ | Resource concentration | μg/mL | | |
| $N$ | Phage sensitive Abi⁻ bacteria | cells/mL | | |
| $Nr$ | Phage resistant Abi⁻ bacteria | cells/mL | | |
| $A$ | Phage sensitive Abi⁺ bacteria | cells/mL | | |
| $Ar$ | Phage resistant Abi⁺ bacteria | cells/mL | | |
| $P$ | Phage sensitive to Abi | pfu/mL | | |
| $Pe$ | Phage resistant to Abi | pfu/mL | | |
| **Parameter** | | | **NCK4/NCK5 + p2** | **IL6/IL7 + P335** |
| $vn, vnr, va, var$ | Maximum growth rates | 1 hour$^{-1}$ | 1.49 ± 0.06 h$^{-1}$, 0.64 ± 0.04 h$^{-1}$, 1.44 ± 0.04 h$^{-1}$, 0.56 ± 0.03 h$^{-1}$ | 1.13 ± 0.02 h$^{-1}$, 1.04 ± 0.02 h$^{-1}$, 1.01 ± 0.01 h$^{-1}$, 0.95 ± 0.04 h$^{-1}$ |
| $\mu nr, \mu rn$ | Mutation rate, $N \leftrightarrow Nr$ | 1e$^{-7}$ per cell/hour | 3.75e$^{-7}$ per cell/hour | <1e$^{-9}$ per cell/hour |
| $\mu ar, \mu ra$ | Mutation rate, $A \leftrightarrow Ar$ | 1e$^{-7}$ per cell/hour | 3.09e$^{-7}$ per cell/hour | <1e$^{-9}$ per cell/hour |
| $\mu pe, \mu ep$ | Mutation rate, $P \leftrightarrow Pe$ | 1e$^{-5}$ per phage/hour | 7.07e$^{-6}$ ± 0.00 per phage/hour | 1.36e$^{-4}$ ± 0.00 per phage/hour |
| $\delta$ | Phage adsorption rate to $N$ or $A$ | 1e$^{-7}$ per mL/hour | 2e$^{-8}$ per mL/hour | 3.6e$^{-7}$ per mL/hour |
| $\beta$ | Phage burst size on $N$ or $A$ | 30 pfu/cell | 24 pfu/cell | 17 pfu/cell |
| $e$ | Conversion efficiency | 1 μg/cell | | |
| $\kappa$ | Monod constant | 1 μg/cell | | |
| $q$ | Abi efficacy ($0 \leq q \leq 1$) | Probability of $P$ infection being aborted by $A$ | | |

In Fig 3 we follow the changes in the densities of bacteria, colony forming units (CFU/mL) and phage, plaque forming units (PFU/mL) over 24 hours projected by our simulations for three situations, first in the absence of resistance or phage that evade Abi (Fig 3A). If abortive infection is completely effective in preventing an infecting phage from replicating, i.e., $q$ = 1.0, the bacterial population increases while that of the phage population declines to extinction. If, however, the probability of abortive infection is $q$ = 0.9, the bacteria are eliminated, and the phage density increases. As can be seen in S3 Fig in S1 File, with the parameters employed for Fig 3A, for Abi to protect a population of bacteria from succumbing to phage, $q$ has to exceed 0.94. Second, we allow for the generation of evolved phage that that can grow on Abi⁺ cells. Even when the efficacy of Abi is complete, $q$ = 1.0, the bacteria are eliminated, and the evolved phage increase in density (Fig 3B). Lastly, we allowed for both evolved phage and resistant bacteria to be generated. The resistant bacteria are anticipated to ascend in all situations where there are phage whether the bacteria are Abi⁺ or Abi⁻ (Fig 3C).

To test the validity of the predictions made from the simulations in Fig 3, we performed 24-hour experiments with NCK4 AbiZ⁻, NCK5 AbiZ⁺ with the phage p2, and IL6 AbiZ⁻ and IL7 AbiZ⁺ with the phage P335. When confronted with phage p2, the NCK5 AbiZ⁺ population replicates as do the phage (Fig 4A). These results are inconsistent with those anticipated from the simulation for the AbiZ⁺ population with an efficacy of $q$ = 1 (Fig 3B), contrary to what is predicted from the model, in the presence of phage there are an abundance of surviving bacteria at 24 hours. The most likely reason for this deviation from the theory is the emergence and ascent of AbiZ⁺ mutants resistant to the phages (S2 Table in S1 File). These resistant bacteria did, however, have a significant fitness cost associated with them as the resistant phenotype

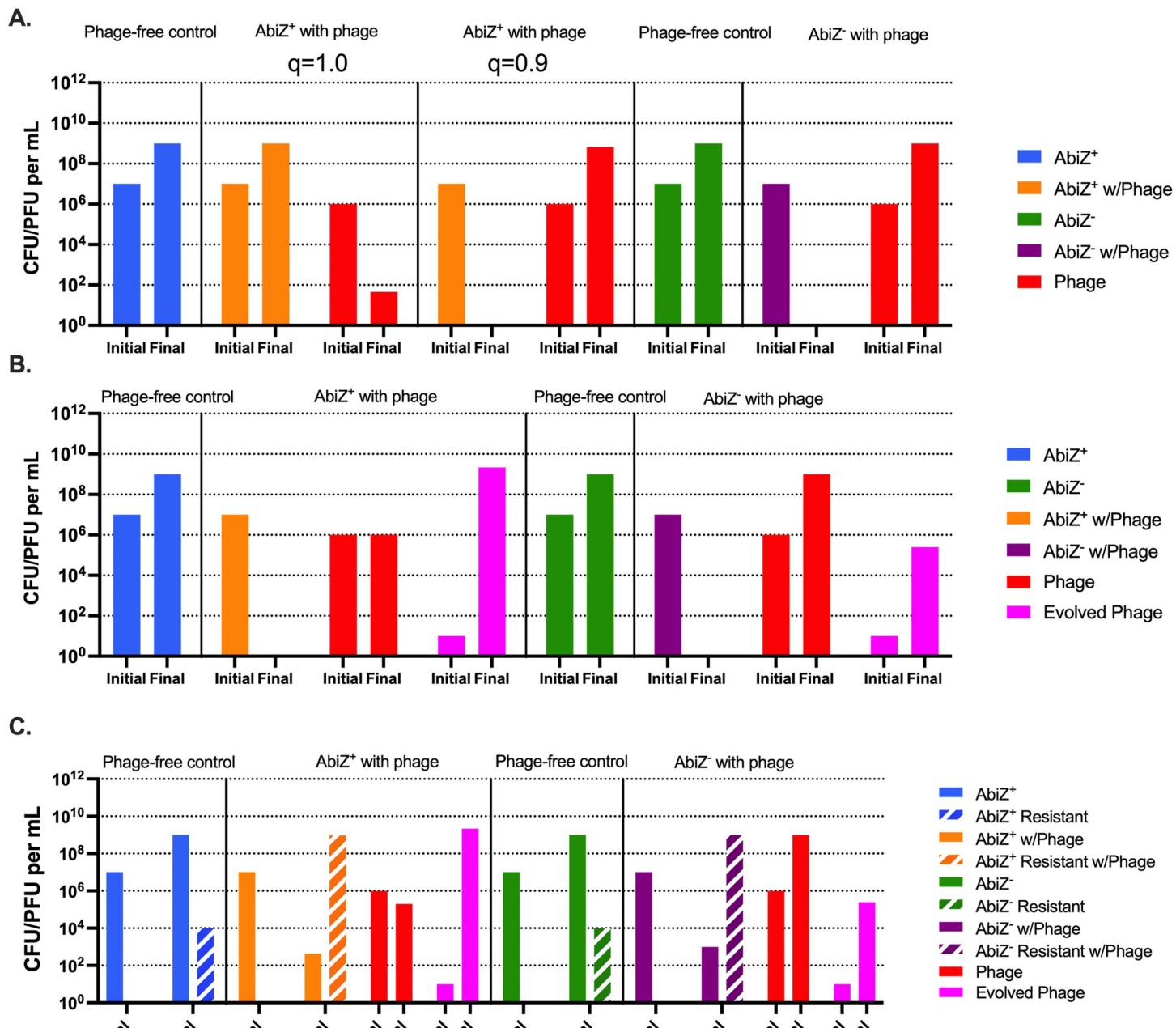

**Fig 3. Computer simulations of the conditions for abortive infection protection against phage. A:** Protection without the emergence of an Abi escape mutant phage and envelope resistance. **B:** Protection with emergence of Abi escape mutant phage and without envelope resistance, $q = 1.0$ **C:** Protection with both evolved phage and envelope resistance emergence, $q = 1.0$. Blue bars represent cells with Abi not confronted by phage. Green bars represent cells lacking Abi which are not confronted by phage. Orange and purple bars represent cells with or without Abi respectively cocultured with phage. Ancestral phage is represented by red bars and evolved phage is represented by pink bars. Envelope resistant mutants of each bacterial population are represented by dashed bars.

coincided with a reduction in growth rate of approximately 60% (Table 1). When these AbiZ$^+$ and AbiZ$^-$ are challenged by p2ev alone (S4A Fig in S1 File), envelope resistance becomes the dominant mechanism accounting for bacterial survival (S2 Table in S1 File).

In the parallel experiments with the AbiZ$^+$ IL7 strain and P335 (Fig 4B), the AbiZ$^+$ population survived and evolved P335 ascended. However, the surviving AbiZ$^+$ IL7 population does

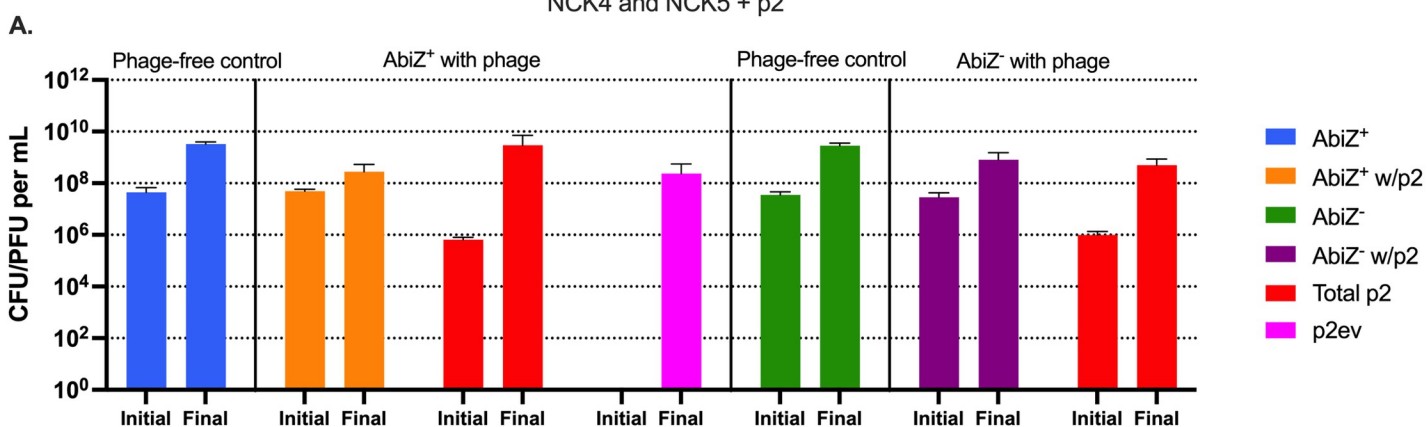

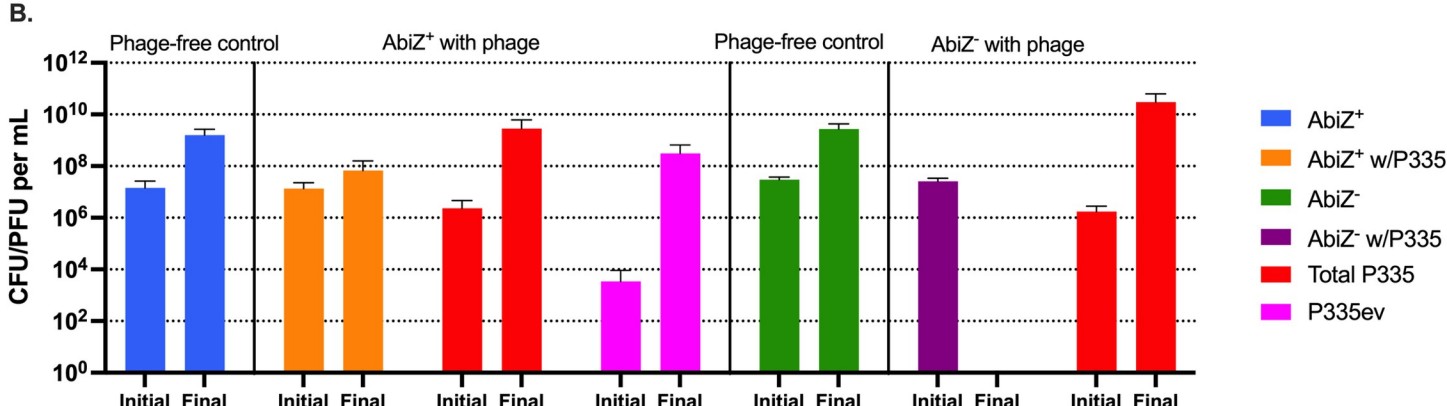

**Fig 4. Conditions for abortive infection protection against p2 or P335 following 24 hours in liquid culture.** Bars represent mean initial (Time = 0hr) and final (Time = 24hr) colony or plaque forming units. **A:** NCK4 and NCK5 + p2, **B:** IL6 and IL7 + P335. Blue bars represent cells with Abi not confronted by phage. Green bars represent cells lacking Abi which are not confronted by phage. Orange and purple bars represent cells with or without Abi respectively cocultured with phage. Total phages are represented by red bars and p2ev/P335ev are represented by pink bars.

not have envelope resistance to the phage (S2 Table in S1 File). AbiZ may protect IL7 from P335 and its evolved mutants, and in the course of 24 hours, the resistant AbiZ+ bacteria have yet to evolve. Consistent with this interpretation is the observation that the AbiZ- IL6 population falls below the limit of detection when confronted by the phage, suggesting that resistance did not ascend to detectable levels in 24 hours. Further evidence for this conclusion is that when AbiZ+ cells are challenged by P335ev alone, no surviving bacteria are detected following 24 hours (S4B Fig in S1 File). Also of note, we did not observe resistant IL6/IL7 colonies during a fluctuation test (Table 1), suggesting the mutation rate is less than the inverse of maximum stationary phase density of IL6 ($\mu < 1 \times 10^{-9}$ per cell/hour).

Although resistance in IL6/IL7 did not emerge in 24 hours, the predictions made by the model and the emergence of resistance in NCK4/NCK5 suggested the emergence of resistance in IL6 and IL7 may occur beyond the 24-hour time point. To test the hypothesis that resistance to P335 would emerge if more time were available, we performed these experiments sampling at 96 hours for the IL6/IL7 P335 system (S5 Fig in S1 File). Following this extended period of

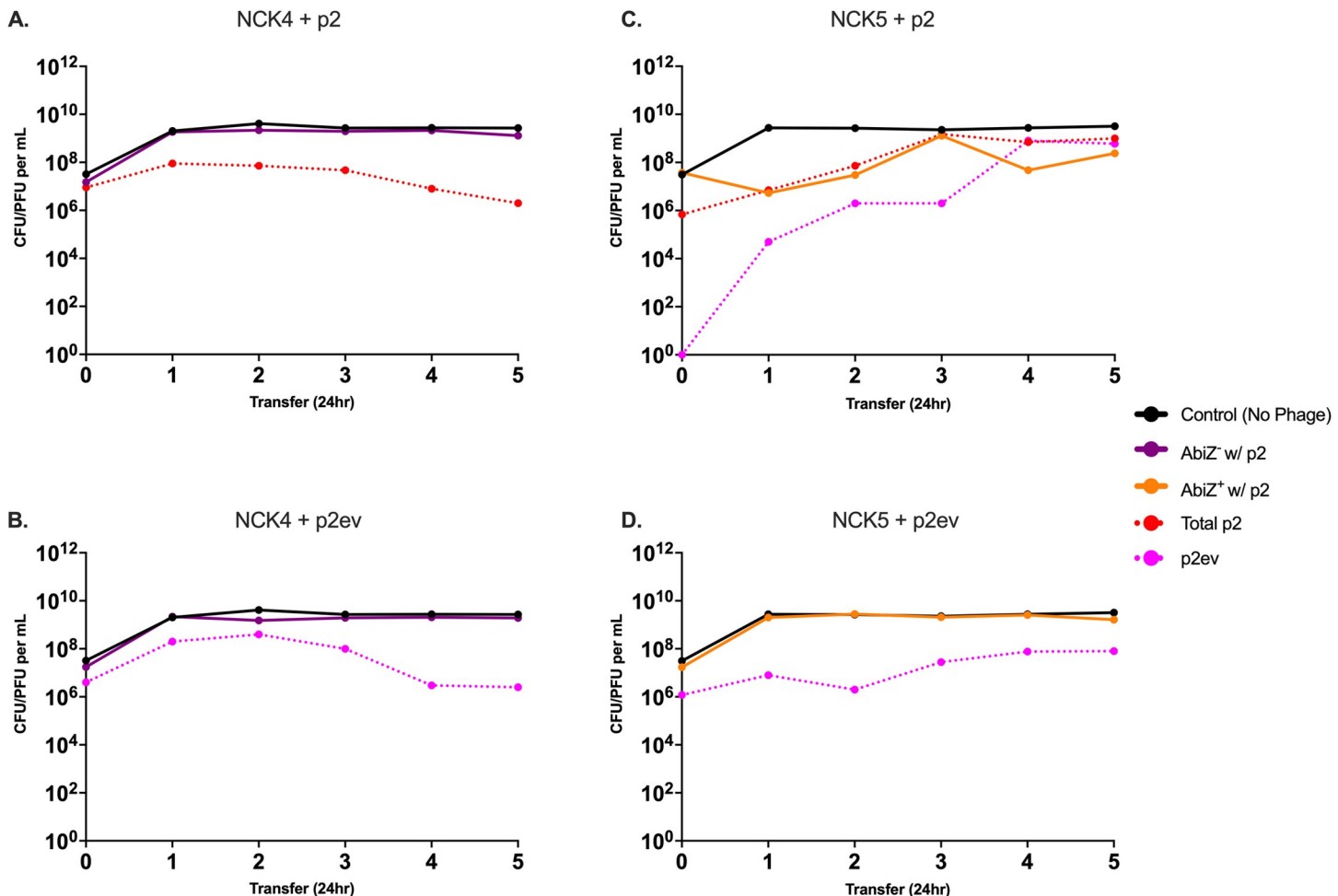

**Fig 5. Serial transfer population dynamics of *L. lactis* NCK4 and NCK5 with p2 and p2ev.** Graphs represent the densities of AbiZ$^+$ and AbiZ$^-$ cells and the densities of the ancestral and evolved phages with which they are confronted. Black lines represent densities of control AbiZ$^+$ and AbiZ$^-$ populations not confronted by phage. **A:** AbiZ$^-$ cells (purple) cultured with p2. Dashed red line represents the total phage (p2 and p2ev). **B:** AbiZ$^-$ cells cultured with p2ev (dashed pink). **C:** AbiZ$^+$ (orange) with p2 and emergence of p2ev. **D:** AbiZ$^+$ cultured with p2ev.

time, both the AbiZ$^+$ and AbiZ$^-$ populations survive the phage and are dominated by envelope resistant cells at a high density (S3 Table in S1 File). These resistant mutants also do not exhibit a clear fitness cost as they did in NCK4/NCK5 (Table 1). These results support the interpretation made about the NCK4/NCK5 and P2 system, that is, the selection for envelope resistance is essential for the maintenance of bacterial populations when confronted by phage and its ensuing Abi escaper.

## Long-term population dynamics: Serial transfer experiments

To further explore the population and evolutionary dynamics of abortive infection and the protection this mechanism provides populations of *L. lactis* from succumbing to phage infection, we performed serial transfer experiments. In the case of AbiZ$^-$, NCK4, when cultured with p2 and p2ev (Fig 5A and 5B) the bacteria survive and are maintained at stationary phase densities across five transfers. When AbiZ$^+$ NCK5 is confronted with p2, despite the emergence of p2ev (Fig 5C) these bacteria are maintained at a high density. A similar result is observed when NCK5 is cultured with a high initial density of p2ev; the bacteria continue to

remain at a high density, suggesting the emergence of envelope resistance is responsible for protection of the bacteria, and not AbiZ alone (Fig 5D). When spot tested, 100% of colonies isolated across the 5 days of transfer are resistant to both the evolved phage and its ancestor, with the exception of NCK5 with the ancestral phage in which AbiZ$^+$ cells lacking envelope resistance survive two transfers before being eliminated by the evolved phage and replaced with an entirely envelope resistant population (S4 Table in S1 File). Notably, in all cases phage are maintained across transfers. These results were consistent across two additional biological replicas shown in S6 Fig in S1 File.

When AbiZ$^-$ IL6 is transferred with P335 and P335ev (Fig 6A and 6B), we observe results different to that observed with its AbiZ$^-$ NCK4 counterpart (Fig 5A and 5B). The bacteria survive at a density above our limit of detection, however, at low densities. AbiZ$^+$ IL7, as in Fig 4, survived the emergence of a P335ev population during the first 24-hour transfer (Fig 6C), but the density of the AbiZ$^+$ population quickly declined and was maintained for the remaining transfers at a density approximately $1 \times 10^2$ CFU/mL. We observe a similar result when IL7 is

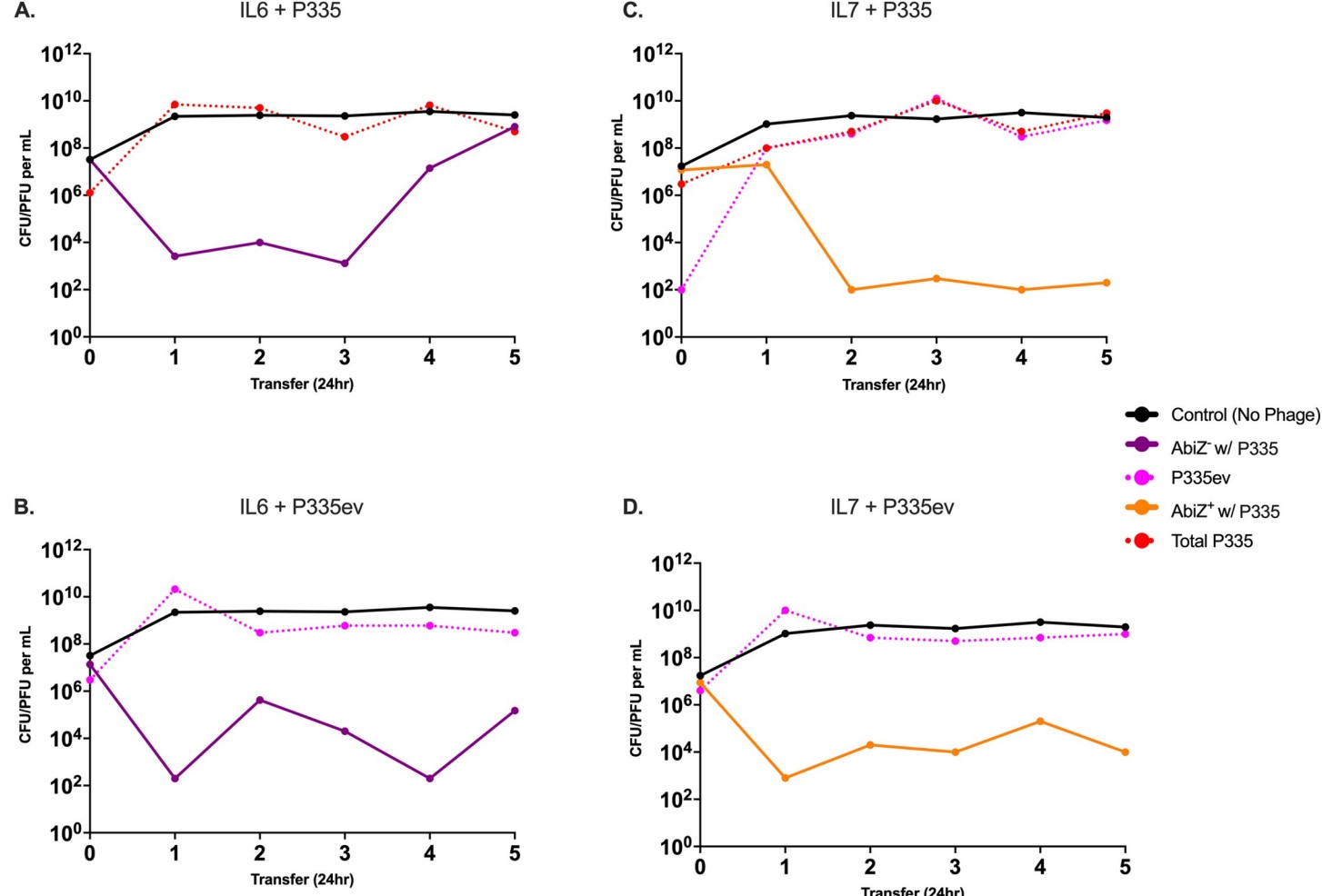

**Fig 6. Serial transfer population dynamics of *L. lactis* IL6 and IL7 with P335 and P335ev.** Graphs represent the densities of AbiZ$^+$ and AbiZ$^-$ cells and the densities of the ancestral and evolved phages with which they are confronted. Black lines represent densities of control AbiZ$^+$ and AbiZ$^-$ populations not confronted by phage. **A:** AbiZ$^-$ cells (purple) cultured with P335. Dashed red line represents the total phage (P335 and P335). **B:** AbiZ$^-$ cells cultured with P335ev (dashed pink). **C:** AbiZ$^+$ (orange) with P335 and emergence of P335ev. **D:** AbiZ$^+$ cultured with P335ev.

transferred with the evolved phage, the bacteria are maintained at a low density across transfers and the phage persist (Fig 6D). These results remain relatively consistent across two biological replicas, with bacterial density at times falling below our limit of detection of $1x10^1$ CFU/mL. In one case with IL7 and P335, the bacteria are eliminated, leaving the phage no host to replicate on leading to phage extinction as well (S6 Fig in S1 File).

Unlike in the NCK4/NCK5 and p2 system, many surviving isolates of IL6 and IL7 tested during the serial transfer experiments did not appear envelope resistant by spot testing. When growth of these isolates was measured by optical density (OD) at an initial MOI of 100 to further test for resistance, we did not observe a clear trend in the maximum OD achieved; however, all final densities were depressed compared to phage free controls and AbiZ$^+$ (IL7) bacteria achieved higher densities overall (S7 Fig in S1 File). With these results failing to show envelope resistance is primarily responsible for the survival of the bacteria in all cases of our IL6/IL7 and P335 system throughout serial transfers, isolates from the 96-hour experiment previously believed to be envelope resistant by spot testing were serially transferred (S8 Fig in S1 File). In all three cases, resistant AbiZ$^+$ and AbiZ$^-$ *L. lactis* survive phage infection and remain at a high density while still allowing for maintenance of phage. At this juncture we do not entirely understand the mechanism for the development of envelope resistant in *L. lactis* IL6/IL7. However, it is clear resistance takes longer than 24 hours to appear. We postulate there is an intermediate population which is not entirely refractory to the phage that may arise during the first 24 hours and persist across serial transfers; however, a secondary mutation is responsible for the development of true envelope resistance observed after 96 hours (S3 Table in S1 File).

## Discussion

In an effort to elucidate the conditions under which abortive infection protects *Lactococcus lactis* populations from extinction by phage, we developed and numerically analyzed the properties of a mathematical model of the population dynamics of bacteria with abortive infection and lytic phage. Our model, an extension of that in Berryhill et al. [21], allows for a population of Abi escape mutant phages and accounts for the evolution of envelope resistance to both the ancestral and evolved phage. To estimate the parameters of this model, and test hypotheses generated from our analysis of their properties, we used two strains of *L. lactis* each of which do or do not harbor the AbiZ abortive infection system, respectively AbiZ$^+$ and AbiZ$^-$. These AbiZ systems cause premature lysis and death of phage-infected cells before the completion of the lytic cycle and the production of phage. These bacteria were also capable of generating mutants with envelope resistance to the phages used in our experiments. Populations of these strains of *L. lactis* were challenged with two phages, p2 and P335, from *L. lactis* phage types 936 and P335, respectively. Both of these phages were able to generate mutants that evade these abortive infection systems.

The results of our theoretical analysis predict that if the Abi system is more than 94% effective in aborting phage infection, Abi can protect populations of bacteria in the absence of the evolution of phage that evade the abortive infection system. While we cannot accurately estimate the efficacy of Abi empirically, our results suggest the efficacy of Abi exceeds the value predicted to be effective, indicating the Abi success rate to be > 94%. Over the short term, AbiZ$^+$ cells prevented the population densities of *L. lactis* from declining in the presence of phage, while AbiZ$^-$ did not. The protection provided by the AbiZ system was, however, lost when mutant phages that evade Abi were generated and ascended. We interpret this to suggest that abortive infection alone is not sufficient to protect populations of *L. lactis* from extinction by lytic phage, but rather is one step in the protection process. We postulate and demonstrate

with *in vitro* experiments that envelope resistance mutants will ascend and be the major mechanism protecting *L. lactis* from phage infection. Notably, despite the evolution of envelope resistance and/or the presence of the AbiZ mechanism, phages are maintained at a high density throughout serial transfers suggesting the envelope resistance developed by *L. lactis* is leaky [29]. Our NCK4/NCK5 and p2 system behaved nearly identically to that were anticipated by our simulations, indicating envelope resistance evolved in relatively short order and was responsible for bacterial survival when escape mutant phages emerge. We observed a more complicated picture in our IL6/IL7 and P335 system. The emergence of envelope resistance through what we postulate to be two or more mutations was essential for maintenance of bacteria population at high densities across serial transfers. This postulation may account for the considerably higher mutation rate in p2 as opposed to P335. Despite differing rates and potentially different number of mutations necessary to develop phage resistance, both systems allow for phage replication at all times, suggesting resistance is very leaky and reversion back to a phage sensitive state occurs at a high rate.

It should be noted that we also identify and characterize with population dynamic experiments the first case of a p2 AbiZ escape-mutant. We also present the first evidence pointing towards a clear mechanism of *Lactococcus lactis* phages ability to escape AbiZ, as both our p2 and P335 escape mutant phages have missense mutations in their major capsid protein.

A great deal of research has been done to understand the molecular and genetic mechanisms of abortive infection, but little consideration has been given to the population dynamics and the conditions under which Abi will protect populations of bacteria from succumbing to phage infection. We set out to address the shortcomings of Abi research and believe we have done so for *L. lac*tis and its phages. However, we see this jointly theoretical and experimental study as only a first step in understanding the contribution of abortive infection to the population and evolutionary biology of *Lactococcus lactis*. A major limitation of this investigation is that it was restricted to the interactions between single phages and bacteria, and only considered two strains of phage. In a natural setting there may well be multiple phages of a number of different types with different receptor sites. It may prove worthwhile to investigate how these dynamics are altered when two or more phages are used and determine if other strains of *L. lactis* and phages would behave differently than those employed here. Our study is also limited to bacteria expressing a single Abi system. AbiZ itself was discovered to coexist on a plasmid with AbiA, another Abi mechanism, and often plasmids code for more than one *L. lactis* abortive infection system or *L. lactis* can harbor multiple plasmids with different Abi systems [15, 18, 30]. It is possible phage escape mutants may be rendered trivial if two separate Abi systems with different mechanisms are at play. To better understand the mechanisms behind phage escape of Abi, further experiments must be done to evaluate the role of the capsid. Similar to our results, many other studies have shown the ability for phages to bypass Abi is solely dependent on mutations in capsid proteins [24, 25]. We consider abortive infection to be both interesting and important from an ecological and evolutionary perspective and look forward to further scientific advances in this area.

## Materials and methods

### (a) Media, phage, and bacteria strains

Bacterial cultures (Table 2) were grown at 30˚C without shaking in M17 Oxoid broth (CM0817B, ThermoFisher) supplemented with 0.5% glucose (M17G) as used in previous studies (17,18). The number of viable colonies (CFU/mL) were counted on M17G Oxoid agar (1.8%). For phage assays, M17G soft agar (0.6%) was used. For all experiments with phage, 10 mM of Calcium Borogluconate and 10 mM of $MgCl_2$ was added to the respective media.

**Table 2. Biological entities used in this study.**

| Biological entity | Acronym in this study | Features | Reference |
|---|---|---|---|
| *Bacteria* | | | |
| *L. lactis* LM0230 | NCK4 | Plasmid-free host for 936- and c2-like phages; cured of prophage and plasmids by nitrosoguanidine and UV treatment; Lac-, R-/M-, plasmid-free transformation recipient | Bouchard et al. 2002; Tanskanen et al. 1990 [32, 33]. |
| | NCK5 | NCK4 + pTRK914 | This study |
| *L. lactis* IL6288 | IL6 | IL6288: prophage-free strain derivative from *L. lactis* ssp. lactis IL1403 | Aucouturier et al 2018 [34] |
| | IL7 | IL6 + pTRK914 | This study |
| *Plasmids* | pTRK914 | pTRK686:*abiZ* | Durmaz et al. 2007 [18] |
| *Phage* | | | |
| p2 | p2 | 936 group | Hill et al. [35] |
| p2ev | p2ev | An evolved version of p2 | This study |
| P335 | P335 | P335 group | Labrie et al. 2008 [36] |
| P335ev | P335ev | An evolved version of P335 | This study |

*Lactococcus lactis* LM0230 (NCK4) strain (Lac-, R-/M-, plasmid-free transformation recipient) and *L. lactis* IL6288 (IL6) strain (Lac-, R-/M-, plasmid-free, prophage free) were obtained from Dr. Rodolphe Barrangou (NCSU). NCK4 and IL6 cells were made electrocompetent and then electroporated with plasmid pTRK914 (pTRK686:abiZ, Cm$^r$) to obtain NCK5 and IL7 (AbiZ$^+$) using methods from [31].

Phage lysates were prepared from single plaques incubated at 30˚C in M17G alongside NCK4 (p2), and NCK5 (p2ev), and IL6 (P335) and IL7 (P335ev). Methods from [18] were used for the isolation of Abi escape mutant phages, phages which can replicate on Abi+ cells, p2ev and P335ev. Chloroform was added to the lysates and the lysates were centrifuged to remove any remaining bacterial cells. Phages p2 and P335 used in this study were obtained from the Félix d'Hérelle Reference Center for Bacterial Viruses, Quebec, Canada, through Dr. Sylvain Moineau.

## (b) Sampling

Bacteria and phage densities were estimated by serial dilution in 0.85% saline solution followed by plating. For phage density, these suspensions were plated at various dilutions on lawns made up of 0.1 mL of overnight M17G-grown cultures of NCK4 or IL6 (about $5 \times 10^8$ cells per mL) and 3 mL of M17G soft agar on top of M17G agar plates. Estimation of evolved phage p2 (p2ev) and P335 (P335ev) densities was performed on NCK5 and IL7 lawns, respectively.

## (c) Testing for and estimating the frequency of phage resistance

Phage resistance was determined using spot testing of bacterial isolates for respective experiments. Phage p2ev and P335ev (10 μL, $>10^8$ plaque-forming units [pfu]/mL) were spotted on agar lawns of these bacteria which were characterized as resistant if no plaques formed.

## (d) Model parameter estimations

Growth rates were estimated using OD in a Bioscreen C and calculated using an R Bioscreen analysis tool found at https://josheclf.shinyapps.io/bioscreen_app. 24-hour overnights of each strain to be tested were diluted in M17G broth to an initial density of approximately $10^5$ cells per mL. 5 technical replicas of each strain were loaded into 100-well plates and grown at 30˚C

(shaking only before measurement) for 24 hours taking OD (600nm) measurements every five minutes. Growth rates were used as an estimation of the population fitness.

Phage burst sizes (β) and adsorption were estimated as described in [37]. Mutation rate was estimated using a fluctuation experiment in which 10 independent cultures of 1E9 *L. lactis* were mixed with 1E9 phage in 3 ml soft agar, the number of colonies produced were estimated and rate of mutation to phage resistance was calculated with the median method described in [38].

## (e) Short-term bacterial growth and phage infection experiments

Approximately $10^7$ CFU/mL of *L. lactis* was added to M17G broth supplemented with Calcium Borogluconate [10mM] and MgCl2 [10mM] and grown for 1 hour at 30˚C without shaking. Following 1 hour, approximately $10^6$ PFU/mL of respective phage was added. Flasks containing bacteria and phage were incubated and bacteria and phage density was sampled every hour for 4 hours and at 7 hours by serial dilution and plating.

## (f) Serial transfer–long term experiments

Approximately $10^7$ CFU/mL of *L. lactis* and $10^6$ PFU/ml phage were added to M17G broth supplemented with Calcium Borogluconate [10mM] and MgCl2 [10mM]. Following 24 hours, a dilution 1/100 was made from the initial culture to a new flask containing fresh M17G media and Calcium Borogluconate [10 mM] and MgCl2 [10 mM]. Cultures continued to be transferred and sampled for phage and bacterial densities every day for 5 days.

## (g) DNA extraction, sequencing, and analysis of sequences

All phages were sequenced using long read technology of Oxford Nanopore Technologies. High titer phage solutions were used to extract phage DNA using Invitrogen's PureLink Viral RNA/DNA extraction kit. After extraction, DNA repair and end-prep occurred, followed by adapter ligation and clean-up using a combination of NEBNext Companion Module for Oxford Nanopore Technologies Ligation Sequencing and Nanopore's Ligation Sequencing Kit. In a 0.2 mL PCR tube, 27 μL of nuclease-free water, 20 μL of sample, 1 μL of DNA CS, 3.5 μL of NEBNext FFPE DNA Repair Buffer, 2 μL of NEBNext FFPE DNA Repair Mix, 3.5 μL Ultra II End-prep reaction buffer, and 3 μL of Ultra II End-prep enzyme mix were added. Tubes were then mixed by flicking and placed onto a thermal cycler and incubated at 20˚C for 5 minutes and 65˚ C for 5 minutes. DNA was then transferred to a new 1.5 mL Eppendorf DNA LoBind tube and 60μL of resuspended AMPure XP beads were added and mixed by flicking. The tubes were then placed on a mixer for 5 minutes and incubated at room temperature. Samples were then pelleted on a magnet and the supernatant was pipetted off. 200 μL of 70% ethanol with nuclease-free water was used to wash the beads without disturbing the pellet twice and then pipetted off each time. Beads were left to dry for 30 seconds and then removed from the magnetic rack and resuspended in 61 μL of nuclease-free water and incubated for 2 minutes at room temperature. The beads were then pelleted on the magnet again and the eluate was removed and retained in a clean 1.5 mL DNA LoBind tube. Next, 60 μL of sample, 25 μL of ligation buffer, 10 μL of NEBNext Quick T4 DNA Ligase, and 5 μL adapter mix were added to a tube and flicked to mix. The reaction was incubated for 10 minutes at room temperature. 40 μL of resuspended AMPure XP beads were added to the reaction and the tube was then incubated on a mixer for 5 minutes at room temperature. The tubes were then pelleted on a magnet rack and the supernatant was removed. The beads were washed and resuspended in 250 μL of Long Fragment Buffer, pelleted using the magnet, and the supernatant was removed twice. The samples were then allowed to dry for 30 seconds and then resuspended in 15 μL of Elution Buffer and incubated for 10 minutes at room temperature. Beads were then

pelleted, and the eluate was removed and retained in a fresh 1.5 mL DNA LoBind tube. These samples could now be stored in the fridge for up to a week before processing. The SpotON flow cell was then loaded in the MinION and sample preparation followed by using the ligation kit. High accuracy base calling with specifications of QScore of at least 7 and reads of at least 20 KB was used for an hour before the process was terminated and the sequences were exported for analysis.

16s rDNA sequence determination was performed by Eurofins (Louisville, Kentucky) to verify *L. lactis* LMN0230/IL6288.

Raw data was analyzed using Geneious Prime de novo assembly and mapping to the RefSeq sequence for p2 (NC042024) and P335 (DQ838728) phages.

Sequences of p2ev and P335ev have been uploaded to GenBank and are available at accession numbers 0R199844 and 0R199843 respectively.

### (h) Efficiency of plaquing (EOP)

The efficiency of plaquing (EOP) was estimated by calculating the ratio of PFU/mL for a phage lysate on an AbiZ$^+$ lawn over an AbiZ$^-$ lawn.

### (i) Mathematical model–simulation methods

Berkeley Madonna$^{TM}$ was used to solve differential equations to create a mass-action model of the experimental system. The infection parameters to use in our simulations were estimated using *L. lactis* LM0230 and IL6288, and phage p2 and p2ev, and P335 and P335ev, respectively.

### (j) Experiments in liquid culture

24-hour overnights of Lactococcus grown at 30˚C in M17G and the p2, p2ev, P335 and P335ev lysates were serially diluted in 0.85% saline to appropriate initial densities and cultured in M17G with Ca Borogluconate [10 mM] and MgCl2 [10 mM]. Final densities of bacteria and phage were measured following 24 hours by serial dilution and plating (96 hours in the case of S5 Fig in S1 File).

### (k) Measurement of IL6/IL7 envelope resistance by OD

Colonies of IL6 and IL7 which survived phage infection during serial transfer experiments were picked and grown overnight at 30˚C in M17G for 24 hours. Overnights were serially diluted in 0.85% saline to approximately $1x10^5$ CFU/mL and grown in the Bioscreen C with approximately $1x10^7$ pfu/mL (MOI = 100) for 24 hours.

## Supporting information

**S1 File. Contains all the supporting tables and figures.**
(PDF)

## Acknowledgments

We thank Dr. Rodolphe Barrangou of North Carolina State University and Dr. Sylvain Moineau of Université Laval for providing us with the phage, bacteria and plasmids used in this study. We thank Andrew Smith for his comments and review of the manuscript.

## Author Contributions

**Conceptualization:** Eduardo Rodríguez-Román, Joshua A. Manuel, Bruce R. Levin.

**Data curation:** Eduardo Rodríguez-Román, Joshua A. Manuel, Bruce R. Levin.

**Formal analysis:** Eduardo Rodríguez-Román, Joshua A. Manuel, Bruce R. Levin.

**Funding acquisition:** Bruce R. Levin.

**Investigation:** Eduardo Rodríguez-Román, Joshua A. Manuel, Bruce R. Levin.

**Methodology:** Eduardo Rodríguez-Román, Joshua A. Manuel, David Goldberg, Bruce R. Levin.

**Project administration:** Bruce R. Levin.

**Software:** Joshua A. Manuel, Bruce R. Levin.

**Supervision:** Bruce R. Levin.

**Validation:** Eduardo Rodríguez-Román, Joshua A. Manuel, Bruce R. Levin.

**Visualization:** Eduardo Rodríguez-Román, Joshua A. Manuel.

**Writing – original draft:** Eduardo Rodríguez-Román, Joshua A. Manuel, Bruce R. Levin.

**Writing – review & editing:** Eduardo Rodríguez-Román, Joshua A. Manuel, David Goldberg, Bruce R. Levin.

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
