## [Decision Letter · Decision Letter 0]

12 Nov 2023

PONE-D-23-26559The contribution of abortive infection to preventing populations of Lactococcus lactis from succumbing to infections with bacteriophagePLOS ONE

Dear Dr. Levin,

Thank you for submitting your manuscript to PLOS ONE. After careful consideration, we feel that it has merit but does not fully meet PLOS ONE’s publication criteria as it currently stands. Therefore, we invite you to submit a revised version of the manuscript that addresses the points raised during the review process.

We look forward to receiving your revised manuscript.

Kind regards,

Konstantinos Papadimitriou, Ph.D.

Academic Editor

PLOS ONE

Journal Requirements:

3. We note you have included a table to which you do not refer in the text of your manuscript. Please ensure that you refer to Table 2 in your text; if accepted, production will need this reference to link the reader to the Table.

Additional Editor Comments:

Dear Authors, I believe that your study is interesting and of importance. However some major revisions are required before the study can be considered for publication.

Reviewers' comments:

Reviewer's Responses to Questions

**Comments to the Author**

1. Is the manuscript technically sound, and do the data support the conclusions?

Reviewer #1: Yes

Reviewer #2: Partly

Reviewer #3: Yes

2. Has the statistical analysis been performed appropriately and rigorously? 

Reviewer #1: Yes

Reviewer #2: I Don't Know

Reviewer #3: Yes

3. Have the authors made all data underlying the findings in their manuscript fully available?

Reviewer #1: Yes

Reviewer #2: No

Reviewer #3: Yes

4. Is the manuscript presented in an intelligible fashion and written in standard English?

Reviewer #1: Yes

Reviewer #2: No

Reviewer #3: Yes

5. Review Comments to the Author

Reviewer #1: The authors describe a model to predict how the phage resistome of Lactococcus lactis drives phage evolution in allowing them to overcome abortive infection systems among other challenges to proliferation. The manuscript is well written, clearly organised and well executed. The approach is thorough and is well articulated. Some minor comments below are suggested to improve the manuscript.

1. In the abstract, the authors should mention which genus of phages are being used and what Abi systems are present in the host that will be discussed in the manuscript.

2. The 936 group of phages have been reclassified as the Skunavirus genus. This should be updated in the manuscript.

3. The motivation for focusing on AbiZ is not clear. It's perfect to do so but needs explanation to the reader.

4. I cannot comment on the mathematical model but the approach seems rational and well considered.

5. The mutation rate is notably higher in p2 than P335. This difference is perhaps worth an explanation in the discussion.

6.

Reviewer #2: line 16: maybe revised

Background information is excessive, can be trimmed.

Material & Methods - should describe the research methods and study design more clearly

Results - the key findings of the study ought to be presented in a succinct fashion, generalities should be avoided.

Line 17-23: too much background information for abstract. the sentence " to combat this.......... and CRISPR-Cas" could be removed.

line 24: species name abbreviated

Introduction:

Line 39: I suggest rewording

line 49-50: rephrase

Line 51-52: what do you want to say here? kindly explain

line 55-56: rephrase

line 57-58: what do the authors want to say here? The concept of population dynamics of Abi explained. line 55-59 should be rephrased

line 63-67: confusing and ambiguous

line 69-73: Why are the study findings discussed at the end of introduction section, instead of outlining the objectives of the study??

Results:

line 75: statistical analysis is missing. Avoid using general statements, merely describing an increase or decrease does not provide clarity, quantitative bent of your results would better convey the message

line 82: can you provide the quantitative data for this. What is meant by proliferate exponentially? Did you calculate growth rate etc.?

What do you mean by and how did you calculate the nutrient-limited densities?

line 86-87: ambiguous statements, provide quantitative data. How did you evaluate this?

Line 91-93: rephrase the sentences. What is meant by "as estimated"?

line 94-98: Increases and decreases must be explained considering statistical differences. General result statements may not be helpful. Provide the quantitative data wherever is possible

Line 100: rephrase. What do you want to say here?

line 129-133: ambiguous... Abi- or (Abi-) are the abbreviations used or N and Nr

Line 151-156: confusing. ? Can the equations be better explained here if that is the intent

Line 159-160: any reference for this?

Line 173: what do you mean by exactly by densities of bacteria and density of phages?

line 183-184: Rephrase

line 207-212: Not comprehensible.

Line 215-216: how did you evaluate this?

line 316: a critical and concise analysis of your results in the light of previously published literature is missing

line 317: follow the same formatting throughout the manuscript

line 321-322: The mathematical model does not appear to account the evolution of envelope resistance? Clarify

line 351-354: rephrase the entire sentence

Line 389: CFU/ml?

Line 391-392: reason and ref?

Line 395: bacterial name should be in italics throughout the manuscript

Line 396: full name?

line 402: ? define Abi escape.

line 431-432: rephrase

line 435: check?

line 439: seems incomplete. short term experiment for?'

Line 444: how?

Line 513-514: how the densities were measured?

Lin3 519: deg centigrade format is not correct

Reviewer #3: The authors embark to elucidate the conditions under which abortive infection confers protection to Lactococcus lactis against phages. They developed and numerically analysed a mathematical model of the population dynamics and interactions of bacteria harbouring abortive infection and phages, followed by an experimental validation. Generally, the article is well written and clear. However, a few things require some attention.

1. The justification for the mathematical model's purpose, which is ‘to provide a framework for a comprehensive consideration of the population dynamics of phage and bacteria with abortive infection’, appears to lack sufficient support. The anticipation of resistance development, a foreseeable outcome based on existing knowledge, diminishes the model's scientific significance considerably. The authors should find a way to justify the importance of the mathematical model.

2. The mathematical model lacks some biologically relevant parameters like phage infection and replication rates in both ABI+ and ABI bacteria. Is there a reason why the authors did not include these parameters?

6. PLOS authors have the option to publish the peer review history of their article (what does this mean?). If published, this will include your full peer review and any attached files.

Reviewer #1: No

Reviewer #2: No

Reviewer #3: No

---

## [Author Response · Author response to Decision Letter 0]

14 Dec 2023

Please see the response to the reviewers document. 

Thank you and have a great holiday season.

---

## [Decision Letter · Decision Letter 1]

30 Jan 2024

The contribution of abortive infection to preventing populations of Lactococcus lactis from succumbing to infections with bacteriophage

PONE-D-23-26559R1

Dear Dr. Levin,

We’re pleased to inform you that your manuscript has been judged scientifically suitable for publication and will be formally accepted for publication once it meets all outstanding technical requirements.

Kind regards,

Konstantinos Papadimitriou, Ph.D.

Academic Editor

PLOS ONE

Additional Editor Comments (optional):

Reviewers' comments:

Reviewer's Responses to Questions

**Comments to the Author**

1. If the authors have adequately addressed your comments raised in a previous round of review and you feel that this manuscript is now acceptable for publication, you may indicate that here to bypass the “Comments to the Author” section, enter your conflict of interest statement in the “Confidential to Editor” section, and submit your "Accept" recommendation.

Reviewer #1: All comments have been addressed

Reviewer #3: All comments have been addressed

2. Is the manuscript technically sound, and do the data support the conclusions?

Reviewer #1: Yes

Reviewer #3: Yes

3. Has the statistical analysis been performed appropriately and rigorously? 

Reviewer #1: Yes

Reviewer #3: Yes

4. Have the authors made all data underlying the findings in their manuscript fully available?

Reviewer #1: Yes

Reviewer #3: Yes

5. Is the manuscript presented in an intelligible fashion and written in standard English?

Reviewer #1: Yes

Reviewer #3: Yes

6. Review Comments to the Author

Reviewer #1: The authors have fully addressed all of my previous queries and suggestions. It is a very nice manuscript that I believe will be of significant relevance to the field.

Reviewer #3: Satisfied revision. All my concerns have been well addressed and I do not have more questions on this work.

7. PLOS authors have the option to publish the peer review history of their article (what does this mean?). If published, this will include your full peer review and any attached files.

Reviewer #1: No

Reviewer #3: No

---

## [Editor Report · Acceptance letter]

21 Mar 2024

PONE-D-23-26559R1 

PLOS ONE

Dear Dr. Levin, 

I'm pleased to inform you that your manuscript has been deemed suitable for publication in PLOS ONE. Congratulations! Your manuscript is now being handed over to our production team.

Kind regards, 

on behalf of

Prof. Konstantinos Papadimitriou 

Academic Editor

PLOS ONE